# Optimization on Secondary Flow and Auxiliary Entrainment Inlets of an Ejector by Using Three-Dimensional Numerical Study

**DOI:** 10.3390/e24091241

**Published:** 2022-09-03

**Authors:** Jia Yan, Jing Jiang, Zheng Wang

**Affiliations:** 1School of Civil Engineering and Architecture, Southwest University of Science and Technology, Mianyang 621010, China; 2School of Artificial Intelligence, Yantai Institute of Technology, Yantai 264003, China

**Keywords:** ejector, three-dimensional, secondary flow, auxiliary entrainment, multi-evaporator refrigeration system

## Abstract

In this paper, by using three-dimensional numerical simulations, the optimization of the cross-sectional area and angle of the secondary flow inlet is first conducted. Then, to further improve the ejector performance, an auxiliary entrainment is proposed and the optimization of the relative position, cross-sectional area and angle of the auxiliary entrainment inlet is accordingly performed by using three-dimensional methods. The results show that: (1) the performance of the ejector with the secondary flow in a vertical direction to the primary flow is slightly better than that in a parallel direction to the primary flow; (2) the effect of the cross-sectional area of the secondary flow has a relatively evident influence on ER, but its effect becomes ignored when the inlet area increases to a certain value; (3) the relative position and axial width of the auxiliary entrainment inlet are important factors influencing ejector performance, and after the optimization of these two geometries, the ejector ER can be increased by 97.7%; and (4) the optimization of the auxiliary entrainment inlet has a substantial effect on the ejector performance as compared to that of the secondary flow inlet. The novelty of this study is that the effect of an auxiliary entrainment on the ejector’s performance is identified by using a three-dimensional numerical simulation for the first time.

## 1. Introduction

With the development of the economy and the rapid growth of the population, the issue of shortage of energy becomes more serious [1]. Therefore, how to reduce the energy consumption of refrigeration systems has become a very popular topic, and the ejector-based refrigeration system is one promising solution since it can usually be driven by waste heat [2]. An ejector is mainly composed of a primary nozzle, a suction chamber, a mixing chamber and a diffuser [1,2,3]. It has some advantages such as a simple structure, no moving parts and low installation and maintenance costs [4,5]. These unique characteristics make the ejector adapted to a wide range of applications, such as solar thermal energy-driven ejector-based air conditioners [6], transcritical carbon dioxide heat pumps [7], proton exchange membrane fuel cells [8,9,10] and ejector-based multi-evaporator refrigeration systems (EMERS) [11].

A lot of research teams have experimentally and numerically investigated the effect of geometries on ejector performance. Wang et al. [12] designed two kinds of auto-tuning area ratio (AR) and nozzle exit position (NXP) ejectors to evaluate the performance of the ejector under different working conditions. The results show that with the increase in primary flow pressure, the optimal AR increases and the optimal NXP decreases, and the performance of the auto-tuning ejector is better than that of the ejector with a fixed structure. Besagni et al. [13] tested the effect of AR with eight different spindle positions on the entrainment ratio (ER) and found that the average COP can be improved by increasing the AR of the ejector. Chen et al. [14] used the adjustable geometry ejector and the ejector with a bypass to improve the ER, and concluded that compared with a general ejector, the maximum increase in ER with an adjustable geometry ejector and an ejector with a bypass are 51.47% and 26.0%, respectively. Yan et al. [15] obtained the optimal converging and diverging angle and length of the primary nozzle with fixed and variable NXPs. Metin et al. [16] conducted a numerical study on the influence of the position of the main nozzle of the ejector and the converging angle of the mixing section. The results indicate that the influence of the converging angle of the mixing section is larger.

However, the majority of the research was based on simplified two-dimensional (2D) axisymmetric models. In order to more accurately study the flow inside the ejector, some researchers established three-dimensional (3D) ejector models. Arun et al. [17] proposed a rectangular ejector and carried out a 3D numerical simulation on it, and optimized the performance of the convergent–divergent nozzle position of the rectangular ejector. Dong et al. [18] used a 3D simulation to study the influence of the length of the mixing chamber on the ejector’s performance. The results show that the entrainment ratio first increased and then decreased with the increase in the length of the mixing chamber, and the ejector had the best performance when the length of the mixing chamber was within the range of 40–80 mm. Mazzelli et al. [19] performed numerical simulations with both 2D and 3D models of supersonic ejectors. Both the 2D and 3D simulation results were consistent with the experimental results under the design conditions; however, under the non-design conditions, only the 3D simulations were consistent with the experimental results. Tao et al. [20] innovatively proposed a three-dimensional momentum exchanger ejector called static pre-cyclonic steam ejector (SCSE). The CFD results revealed that the SCSE with a smaller axial size had a better entrainment ratio compared with traditional ejectors. Ś Mierciew et al. [21] performed a simulation study with 3D models of the ejector, and they obtained the key geometric parameters of the ejector.

Apart from the above-mentioned studies, an ejector-based multi-evaporator refrigeration system (EMERS) was recently proposed and studied by some researchers [22,23,24]. The schematic and P–h diagram of an EMERS are displayed in Figure 1a,b, respectively [22]. For such system, Wen et al. [23,24] studied the effect of key geometric dimensions, operating conditions and cooling loads on ejector performances with 2D simulation methods. Li et al. [22,25] investigated the effect of crucial ejector geometries on ejector performances of such system by using a 2D numerical study. Also using a 2D numerical study, Lin et al. [26] investigated the adaption of an adjustable ejector with variable cooling loads and the characteristics of the ejector pressure recovery of an EMERS. The results indicate that the adjustable ejector using spindle to adjust the throat area of the primary nozzle is an effective way to maintain system stability. Liu et al. [27] analyzed the efficiency of an improved dual ejector in a transcritical CO_2_ refrigeration cycle by using a 2D numerical study.

Previous studies relevant to EMERS indicated that the use of ejectors can improve the energy efficiency of the refrigeration system; however, almost all of them were carried out by using a 2D numerical simulation, thus, multiple secondary inlets as shown in Figure 2 and multiple auxiliary entrainment pipe as displayed in Figure 3 cannot be addressed if still using 2D simulation methods. Therefore, the purpose of this study is to establish 3D simulation ejector models, and to optimize the secondary flow inlet and the auxiliary entrainment inlet of the ejector in an EMERS.

The rest of this paper consists of the following sections:Section 2 describes the experimental system;Section 3 introduces the 3D CFD modelling;Section 4 presents the numerical method and validation of the CFD model;Section 5 reveals the optimization of the secondary flow and auxiliary entrainment inlets of the ejector by using a 3D numerical simulation.

## 2. System Description

The experimental system is a multi-evaporator refrigeration system based on a two-stage ejector, and its schematic is shown in Figure 4. The main components of the system consist of a compressor, a condenser, three electronic expansion valves (EEVs), three electric evaporators and two ejectors. The working medium is R134a. The photograph of the experimental rig can be referred to in our previous work [23].

In this system, Evaporator 1, Evaporator 2 and Evaporator 3 are simulated as freezer, refrigerator and air-conditioner, respectively. Table 1 is the evaporating temperature and designed cooling load of the three evaporators. The system can generate two working conditions by using the following procedures: (1) EEV1, EEV2 and V5 are opened and EEV_3_ is closed, the fluid coming from the refrigerator goes to the primary nozzle of Ejector 1 and entrains the fluid out of the freezer chamber, and this is the single-stage ejector’s working condition for the refrigerating–freezing mode; (2) EEVs 1, 2, and 3 are opened and V5 is closed, and this can produce the two-stage ejector’s working condition. The single-stage ejector’s working condition is selected for the rest of the study.

Meanwhile, high precision pressure transmitters with resistance strain and temperature sensors of PT1000 are used for pressure and temperature measurement in this experiment. Considering the flow rate range, elliptical gear flowmeters are used to measure the volume flow rate and details of the measurement sensors are shown in Table 2.

As for the uncertainty of the measurement, it mainly considers the precision limits of the experimental results caused by the accuracy of the sensors. Uncertainty for temperature, pressure and flow rate can be written as below:(1)$$U_{I}=\frac{P_{\left(I\right)}}{I}$$
where,
*U_I_*: Uncertainty for temperature, pressure and flow rate;*P*_(*I*)_: Uncertainty due to accuracy;*I:* Measured temperature, pressure and flow rate.

The uncertainty for ER is as follows:(2)$${\left(U_{ER}\right)}^{2}={\left(\frac{P_{\left(ER\right)}}{ER}\right)}^{2}={\left(\frac{P_{\left(M_{p}\right)}}{M_{p}}\right)}^{2}+{\left(\frac{P_{\left(M_{s}\right)}}{M_{s}}\right)}^{2}$$
where,
$U_{ER}=\frac{P_{\left(ER\right)}}{ER}$: Uncertainty for ER;$\frac{P_{\left(M_{p}\right)}}{M_{p}}$: Uncertainty for primary flow rate;$\frac{P_{\left(M_{s}\right)}}{M_{s}}$: Uncertainty for secondary flow rate.

After calculation, the uncertainty of seven parameters (T1, T2, P1, P2, M1, M2, and ER) were obtained with values of ±6.25%, ±1.04%, ±1.63%, ±4.76%, ±4.40%, ±3.60% and ±5.68%, respectively.

## 3. CFD Modelling

### 3.1. Initial Ejector Geometries

The schematic of the symmetry plane of a classical three-dimensional ejector is shown in Figure 5, in which the ejector is divided into five key parts: primary nozzle, suction chamber, constant-pressure mixing chamber, constant-area mixing chamber and diffuser. The ejector performance is assessed by ER defined as:(3)$$ER=\frac{\overset{•}{M_{S}}}{\overset{•}{M_{P}}}$$
where $\overset{•}{M_{P}}$ is the mass flow rate of primary flow (g·s^−1^), and $\overset{•}{M_{S}}$ is the mass flow rate of secondary flow (g·s^−1^). According to [28,29,30], the designed geometrical parameters of the initial ejector are listed in Table 3.

### 3.2. Governing Equations

For the flow and mixing process in the ejector, the assumptions are given as follows: the flow in the ejector is stable, the inner wall of the ejector is insulated and the fluid is an ideal gas [25]. Thus, governing equations for mass, momentum and energy conservation can be derived as below:

Continuity equation:(4)$$\frac{\partial \left({\rho u}_{i}\right)}{\partial x_{i}}=0$$

Momentum equation:(5)$$\frac{\partial \left({\rho u}_{j}u_{i}\right)}{\partial x_{j}}=\frac{\partial \tau_{\mathrm{ij}}}{\partial x_{j}}-\frac{\partial P}{\partial x_{i}}$$

Energy equation:(6)$$\frac{\partial \left(u_{i\;}\left(\rho E+P\right)\right)}{\partial x_{i}}=\dot{\nabla }\cdot \left(\alpha_{\mathrm{eff}}\frac{\partial T}{\partial x_{i}}+u_{i}\tau_{\mathrm{ij}}\right)$$
where stress tensor $\tau_{\mathrm{ij}}$ is:(7)$$\tau_{\mathrm{ij}}=\mu_{\mathrm{eff}}\left(\frac{\partial u_{i}}{\partial x_{j}}+\frac{\partial u_{j}}{\partial x_{i}}-\frac{2}{3}\frac{\partial u_{k}}{\partial x_{k}}\delta_{\mathrm{ij}}\right)$$
where i, j and k are space vector directions, u is velocity vector (m·s^−1^), P is pressure (Pa), T is temperature (K), E is total energy (J·kg^−1^), ρ is mixture density (kg·m^−3^), $\alpha_{\mathrm{eff}}$ is effective thermal conductive (W·m^−1^·K^−1^), $\delta_{\mathrm{ij}}$ is Kronecker delta function,${\;\mu }_{\mathrm{eff}}$ is effective dynamic viscosity (N·s·m^−2^).

## 4. Numerical Method and Validation of CFD Model

### 4.1. Numerical Method

Fluent 19.0 is used to solve the governing equations, and the 3D solver is applied to the ejector model (as shown in Figure 6). The nonlinear deviation governing equations are linearized from a series of algebraic equations [23]. In addition, a pressure-based standard order upwind scheme is adopted. The SIMPLE algorithm is used to obtain the pressure field. The standard k-ε is selected as the turbulence model, and the standard wall function is used near the wall. The physical parameters of the working fluid R134a are obtained from the NIST database. Although R134a has been gradually phased out in many western countries, it can still be used for a long time in many developing countries. The boundary conditions are shown in Table 4, in which the primary inlet and secondary inlet are set as the pressure inlets, the outlet is set as the pressure outlet, and 10 K of overheating is applied for the fluids at both inlets. The relaxation factor for pressure is set as 0.1 and those for other variables are set as 0.3. The residual convergence limit of the energy equation is set as less than 10^−6^, and the residual convergence limit of the other equations is set as less than 10^−5^.

To verify the grid independence of the model, three grid densities (coarse density grids of 294,275, medium density grids of 544,473 and fine grids of 886,798) are used to calculate the axial static pressure. As shown in Figure 7, Point A (46, 0, 0) which is at the central position of the ejector throat outlet, is selected as the verification checkpoint, because its pressure and velocity are sensitive to the mesh size. The verification results are shown in Table 5. It can be seen that the pressure error decreases with the increase in the number of grids. When the number of grids increases from 544,473 to 886,798, the pressure error of Point A is only 1.48%. Furthermore, the velocity errors at Point A are all less than 1%, which can be ignored. Therefore, in order to guarantee the simulation quality and save calculation time, the medium density grids of 544,473 are eventually used for the following simulation.

### 4.2. Validation of CFD Model

Table 6 shows 12 sets of experimental operating conditions used for the validation of the CFD model under three groups. The validation results obtained are shown in Figure 8. For group 1, the average and maximum deviations of the CFD simulation and experimental results are 8.64% and 11.47%, respectively; the average and maximum deviations of group 2 are 7.81% and 10.18%, respectively; and the average and maximum deviations of group 3 are 7.55% and 10.19%, respectively. The maximum deviation between the simulation results and the experimental results are within 12%. The results of the three-dimensional simulation of the ejector are very close to the experimental results. Therefore, the 3D ejector model can be used for the following simulation.

## 5. Results and Discussion

### 5.1. Optimization of Cross-Sectional Area of Secondary Flow Inlet under Parallel Direction of the Primary Flow Inlet

The influence of the number of secondary flow inlet holes (or the cross-sectional area of the secondary flow inlets) on the ER is presented in Figure 9. To be specific, the position of the secondary flow inlet is above the primary flow inlet in the case of a single secondary flow hole; when the number of secondary flow holes is two, the angle between the two secondary flow holes is 180°; and the angle between each secondary flow hole is 120° for the three secondary flow holes, which are 90°, 72°, 60°, 51.4° and 45° for four to eight secondary flow holes, respectively.

The variation of entrainment ratio with the number of secondary flow inlet holes (HN) is illustrated in Figure 10. It can be found that ER increases along with the increase in HN, or the minimum value of ER is 0.3952 when the HN is one, and the maximum value of ER is 0.4319 when it is 8. However, when the HN changes from one to two, the ER increases from 0.3952 to 0.4181, with an obvious growth rate of 5.79%. When the HN changes from two to four, the ER rises from 0.4181 to 0.4270 with an improvement rate of 2.13%. As the HN increases from four to six, the ER increases from 0.4270 to 0.4312, with an increase rate of 0.98%. As the HN further increases from six to eight, the ER improves from 0.4312 to 0.4319, with an increase rate of only 0.16%. That is to say, the increase in entrainment ratio becomes less and less with the increase in HN, and in particular the increase in ER is almost unchanged when the HN is beyond 6. Therefore, the influence of the cross-sectional area of the secondary flow inlet on ER can be ignored when the HN is greater than six. The phenomenon of the increase in ER with HN can be explained with the Mach number distribution along the central symmetry plane (*y* = 0) as illustrated in Figure 11a–e. From Figure 11a,b, it can be seen that the secondary flow cannot completely reach the lower part of the suction chamber with one hole as it can when the suction chamber has two holes, namely, only the upper part of the suction chamber produces a good entrainment effect. Therefore, the entrainment ratio with one hole of the secondary flow inlet is evidently lower than that of two symmetric holes. In Figure 11c–e, the Mach number distribution is almost equal when the secondary flow has five to seven inlet holes, or the difference from mixing the two flows can be ignored, so that the difference of ER is nearly negligible. That is to say, the influence of the secondary flow inlet cross-sectional area can be ignored when the number of secondary flow inlet holes is equal to or greater than six. Therefore, the optimal number of secondary flow inlet holes can be selected as six.

Based on the above results, when the direction of the secondary flow is parallel to the primary flow, the inlet holes or the inlet area of the secondary flow have a certain influence on the entrainment ratio, but said influence becomes ignored when the holes or the area increase to a certain value.

### 5.2. Optimization of Secondary Flow Inlet in Vertical Direction of Primary Flow Inlet

The optimization of the angle and the cross-sectional area of the secondary flow inlet on the performance of the ejector is conducted in this section when the secondary flow inlet is in the vertical direction of the primary flow inlet. As shown in Figure 12, the secondary flow inlet is located on the suction chamber in the vertical direction of the primary flow inlet. In addition, the secondary flow inlet is in a circle with a diameter of 5 mm and its relative position from the left end of the suction chamber is set as *l*_0_.

#### 5.2.1. Angle Optimization of Secondary Flow Inlet

In this section, when the relative position of the secondary flow inlet is at *l*_0_ = 0, the study of the impact of the secondary flow inlet angle (*θ*, as shown in Figure 12) on the ER is carried out. The results are shown in Figure 13. When *θ* increases from 30° to 150° with an interval of 30°, the ER decreases from the maximum value of 0.4345 (30°) to the minimum value of 0.4320 (150°). Thus, the optimal entrance angle should be 30°. In order to further find out the optimal secondary flow inlet angle, the simulation with a smaller interval of secondary flow entrance angle of 5° is next conducted when the *θ* varies from 20° to 40°. It can be found that the ER first increases from 0.4337 (20°) to the maximum value of 0.4345 (30°), and then decreases to 0.4342 (40°), so the optimal angle of the secondary flow inlet is 30°. However, the deviation of ER is only 0.6% when the *θ* increases from 20° to 150°, which indicates that the change of the angle of the secondary flow inlet has an insignificant influence on the performance of the ejector.

#### 5.2.2. Optimization of Cross-Sectional Area of Secondary Flow Inlet

As can be seen from Section 5.2.1, the influence of the secondary flow inlet angle on the ER is so weak that it can almost be ignored. In this part, the study is performed to identify the effect of the number of secondary flow inlet holes or the cross-sectional area of secondary flow inlets on the ER, when the diameter and angle of the secondary flow inlet are 5 mm and 90°, respectively. The number and distribution of the secondary flow inlet holes in the plane of *x* = 0 are illustrated in Figure 14. The angles between holes with different numbers of holes are the same as those in Section 5.1.

Figure 15 shows the variation of ER with the number of secondary flow inlet holes (HN). It reveals that the ER increases with the increase in HN. The minimum value of ER is 0.3978 when the number of holes is one and the maximum value is 0.4337 when the number of holes is eight. In addition, the range of ER is 0.3978–0.4211 when the holes increase from one to two, which means that ER increases by 5.86%; and the range of ER is 0.4211–0.4315 with holes varying from two to four, and the ER therefore increases by 2.47%; and the range of ER is 0.4315–0.4335 when holes rise from four to six, which leads to an increase in ER of 0.46%; and the range of ER is 0.4335–0.4337 when the holes increase from six to eight, which means that ER increases by only 0.05%. Thus, as the number of holes increases, the growth rate of ER decreases. In comparison with Section 5.1, it can be found that for the same number of holes, the ER of ejector when the secondary flow entering in the vertical direction is higher than that in the parallel direction of the primary flow. It can be concluded that when the direction of the secondary flow is perpendicular to the direction of the primary flow, the performance of the ejector is slightly better than that of the secondary flow entering in parallel to the primary flow.

When the secondary flow is perpendicular to the direction of the primary flow, to summarize, the change of angle of the secondary flow inlet has negligible influence on ER, while the change of area of the secondary flow inlet has relatively evident influence on ER; however, its influence becomes ignored when the holes or area increase to a certain value. The influence characteristic is similar to that found in a parallel direction, but the performance of the ejector in a vertical direction is slightly better than that in a parallel direction.

### 5.3. Effect of Auxiliary Entrainment on Ejector Performance

Based on the optimal secondary flow inlet geometries determined in Section 5.2, the effect of the auxiliary entrainment on the ejector’s performance is presented in this section. As mentioned in Section 1, Li et al. [25] proposed a two-dimensional study on the effect of auxiliary entrainment; however, they used a 2D simulation, so they cannot address the case of a circular hole being connected to the diffuser as shown in Figure 16. In addition, they only analyzed the ejector working for the air conditioning–freezing mode rather than the refrigerating–freezing mode. Furthermore, they did not mention the effect of the axial width of the auxiliary entrainment inlet on the ejector’s performance.

#### 5.3.1. Optimization of Relative Position of Auxiliary Entrainment Inlet

Initially, the auxiliary entrainment inlet is connected to the secondary flow inlet with a ring shape in the radial direction as shown in Figure 17. In addition, the ring width of the auxiliary entrainment inlet is set as 3 mm, and the distance between the left end of the auxiliary entrainment inlet and the leftmost end of the ejector is set as 46 mm, that is to say, the left end of the auxiliary entrainment inlet is initially located at the entrance of the diffuser. The simulation is then carried out by moving the auxiliary entrainment inlet from the initial position to the outlet of the diffuser with an interval of 1 mm.

Figure 18 shows the effect of the relative position of the center of the auxiliary entrainment inlet on the ER. It can be found that the value of the ER at the initial position (the center of the hole is 47.5 mm) is 0.641. As the position of the auxiliary entrainment inlet is moved, the ER increases to the peak value of 0.6455 (the center of the hole is 49.5 mm) and then gradually decreases. When 54.5 mm represents the center position of the inlet, the ER decreases to 0.3063, which is less than the value of 0.4327 without the auxiliary entrainment as obtained in Section 5.2. Hence, it can be concluded that the optimal auxiliary entrainment inlet position is 49.5 mm, and the ER at the optimal auxiliary entrainment inlet position increases by 49.17% compared with that without auxiliary entrainment.

The substantial improvement of ER with auxiliary entrainment can be analyzed with the Mach number distribution as displayed in Figure 19a,b. As can be seen from Figure 19a, when there is no auxiliary entrainment, the velocity of the mixed fluid at the entrance of the diffuser section increases sharply, which leads to a decrease in the static pressure. Therefore, if there is an auxiliary entrainment deployed in this section, the secondary fluid may be more induced, as can be seen in Figure 19b. After the auxiliary entrainment is added to the left part of the diffuser, the auxiliary entrainment fluid is fully mixed with the mixed fluid, and thus, a good entrainment effect is generated.

#### 5.3.2. Optimization of Auxiliary Entrainment Inlet Area

According to the results of Section 5.3.1, the ER can reach 0.6455 when the auxiliary entrainment inlet is opened in a ring around the wall of the diffuser with a distance from the diffuser’s left end of 15 mm. In the following study, the radial width of the auxiliary entrainment inlet ring is changed, whilst the auxiliary entrainment inlet’s relative position and its axial width are kept constant. The number of auxiliary entrainment inlet rectangles increases from one to five (the auxiliary entrainment entrance is shown in Figure 3), and the distribution of auxiliary entrainment inlet rectangles with a calculated angle between each rectangle is similar to that found in Figure 14. The variation of the ER with the number of auxiliary entrainment inlet rectangles is displayed in Figure 20. The results are as follows: the ER increases with the increase in the number of rectangles, the minimum value of the ER is 0.4313 when the number of rectangles is one, which is even less than the optimal ER of 0.4327 obtained in Section 5.2 without auxiliary entrainment. The reason may be explained with the Mach number contour of Figure 21; when the number of rectangles increases to two, the ER increases to 0.5085, indicating that the auxiliary entrainment produces a good effect. As the number of rectangles increases to five, the maximum value of the ER increases to 0.6234, and the maximum deviation of the ER is 44.5% when the number of rectangles increases from one to five. In summary, the number of auxiliary entrainment inlet rectangles has evident influence on the ER. Thus, under the condition of a constant axial width of auxiliary entrainment inlet rectangles, the ejector offers a better performance if the number of auxiliary entrainment inlet rectangles is greater, or the ejector has a better performance if the auxiliary entrainment inlet is in a ring around the diffuser section as presented in Figure 17.

With the optimum radial ring of the auxiliary entrainment inlet, the simulation is next conducted by changing the axial width of the auxiliary entrainment inlet ring. The axial width of the auxiliary entrainment inlet ring varies from 2 mm to 21 mm with an interval of 1 mm, and the center position of the auxiliary entrainment inlet ring remains unchanged. As shown in Figure 22, with the increase in the axial width, ER first increases and then reaches a maximum value of 0.859 when the ring width increases to 15 mm. Then, the ER decreases and drops to 0.806 when the axial width gets to 21 mm. The maximum deviation of the ER is 42.11%, and thus, the change of axial width in the auxiliary entrainment inlet ring has great influence on the entrainment ratio.

#### 5.3.3. Optimization of the Auxiliary Entrainment Inlet Angle

Based on the optimal central position and axial width of the auxiliary entrainment inlet ring obtained in the previous two sections, the influence of the auxiliary entrainment inlet ring angle on the entrainment ratio is then studied. The results are shown in Figure 23. It can be seen that, as the auxiliary entrainment inlet ring angle increases, the ER first increases and reaches the maximum value of 0.867 at 60°, and then gradually decreases and reaches the minimum value of 0.834 at 120°. Thus, it can be concluded that the optimal angle of the auxiliary entrainment inlet ring is 60°, which is different from the result obtained by Li et al. [25]. Moreover, the maximum deviation of ER is 3.94% when the angle changes from 30° to 120°. This indicates that the auxiliary entrainment inlet ring angle has little influence on the ejector’s performance.

To summarize, the relative position of the auxiliary entrainment inlet has a significant influence on the ER, and it is at 15 mm of the inlet of the diffuser section that it provides optimal ER. Moreover, the axial and radial widths of the auxiliary entrainment inlet ring have a relatively apparent impact on the ER, and the ring with 15 mm of axial width is the best. Furthermore, the angle of the auxiliary entraining inlet has little influence on the performance of the ejector, and the optimal angle of the auxiliary entrainment inlet ring is 60°.

## 6. Conclusions

In this paper, a 3D numerical simulation on the optimization of the secondary and auxiliary entrainment flow inlets of an ejector equipped in a multi-evaporator refrigeration system is carried out. The main findings are as follows:The cross-sectional area of the secondary flow inlet has a relatively apparent influence on the entrainment ratio; as the cross-sectional area of the secondary flow inlet increases, the growth rate of the ER in both parallel and vertical directions exceeds 9%, but its influence becomes ignored when the inlet area increases to a certain value.The performance of the ejector with a secondary flow in the vertical direction of the primary flow is slightly better than that in the parallel direction of the primary flow.With auxiliary entrainment, the relative position of an auxiliary entrainment inlet ring has a great influence on the ER, and it is at 15 mm from the inlet of the diffuser section that it provides optimal ER, which increases by 49.2% compared to that without auxiliary entrainment.The axial and radial widths of the auxiliary entrainment inlet ring have substantial influence on the ER, as based on the optimal relative position of the auxiliary entrainment inlet ring, the ER with the optimal axial and radial widths improves by 48.5%.

This study involves 3D simulation to improv ejector performance, which has a practical significance in enhancing the performance of multi-evaporator refrigeration systems. Further studies relevant to two-stage ejectors with auxiliary entrainment will be carried out in the near future.

## Figures and Tables

**Figure 1 entropy-24-01241-f001:**
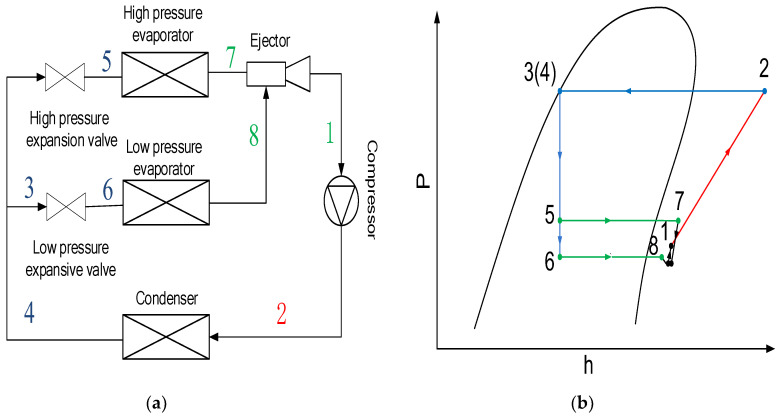
Schematic and P–h diagram of an EMERS: (**a**) Schematic; (**b**) P–h diagram.

**Figure 2 entropy-24-01241-f002:**
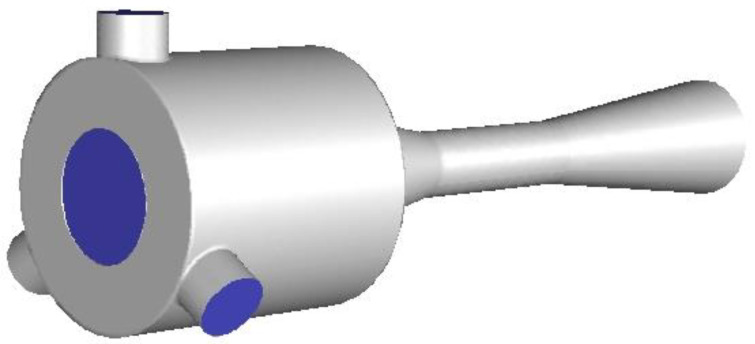
An ejector with multiple secondary flow inlets.

**Figure 3 entropy-24-01241-f003:**
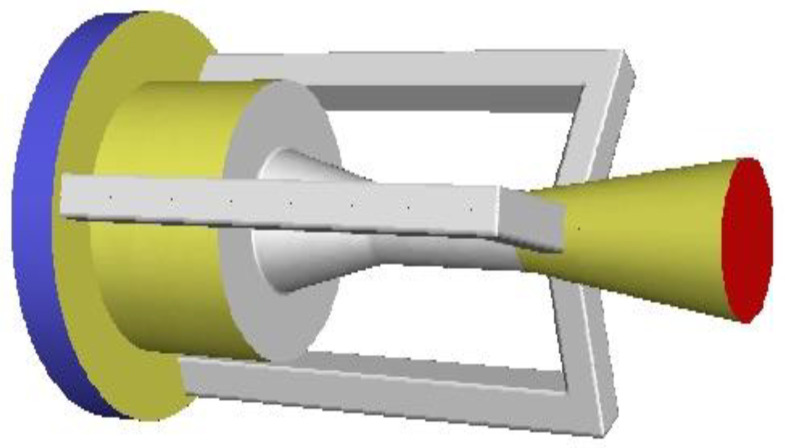
An ejector with multiple auxiliary entrainment pipes.

**Figure 4 entropy-24-01241-f004:**
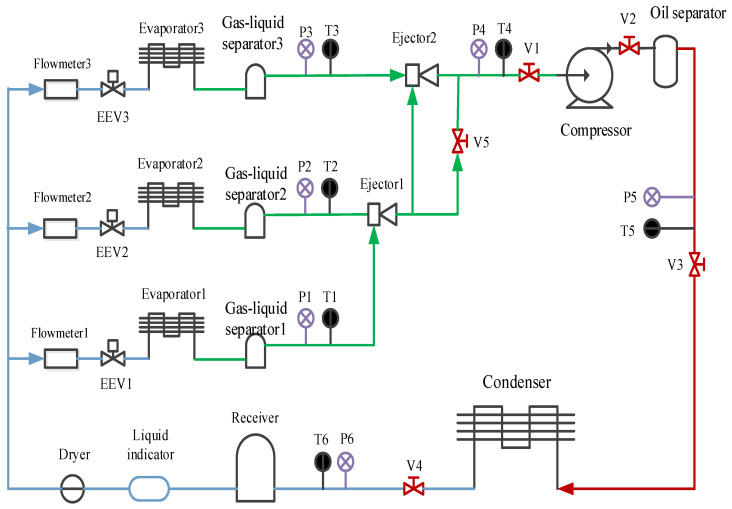
Schematic of the experimental system.

**Figure 5 entropy-24-01241-f005:**
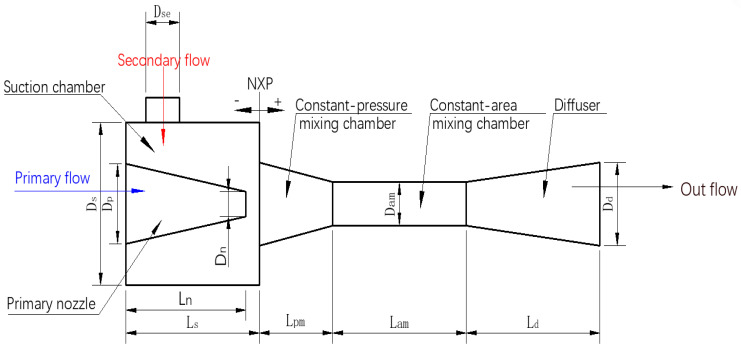
Schematic of the symmetry plane of a classical three-dimensional ejector.

**Figure 6 entropy-24-01241-f006:**
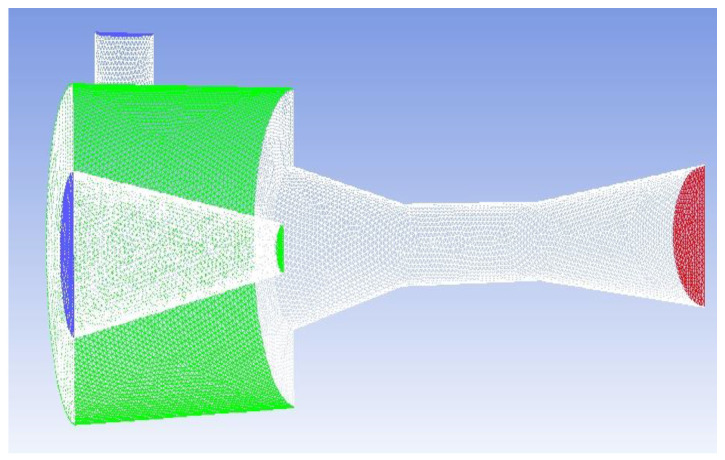
3D grid distribution of the ejector.

**Figure 7 entropy-24-01241-f007:**
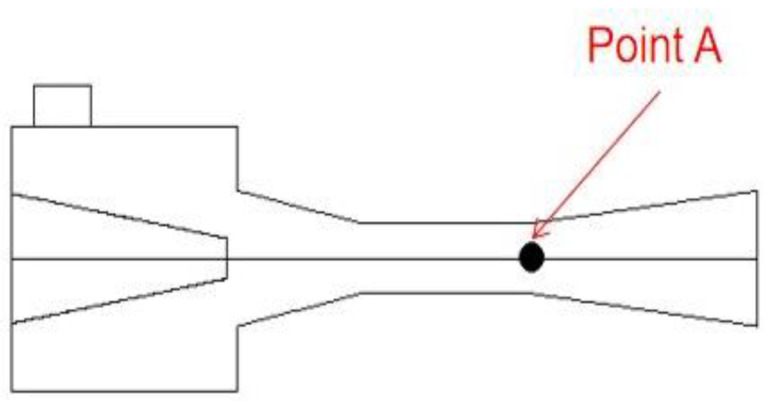
Location of Point A in the central symmetry plane (*y* = 0).

**Figure 8 entropy-24-01241-f008:**
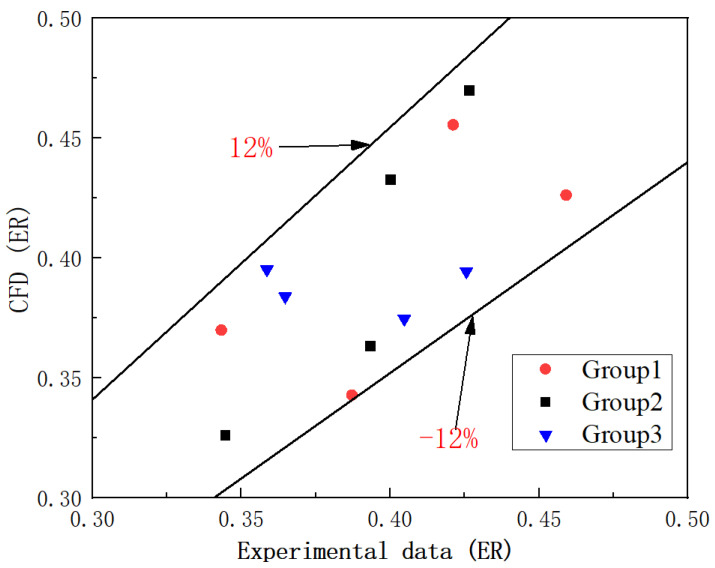
The comparison of ER between the CFD model and experimental data.

**Figure 9 entropy-24-01241-f009:**
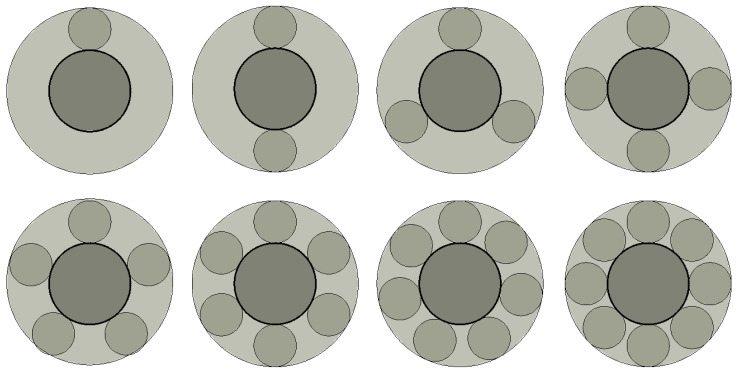
Number and distribution of secondary flow holes when two flows are in parallel direction.

**Figure 10 entropy-24-01241-f010:**
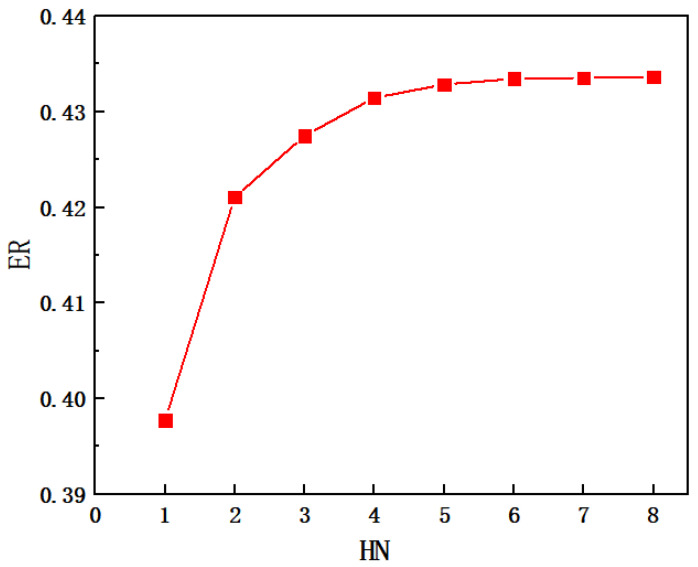
The variation of ER with the number of secondary flow holes (HN).

**Figure 11 entropy-24-01241-f011:**
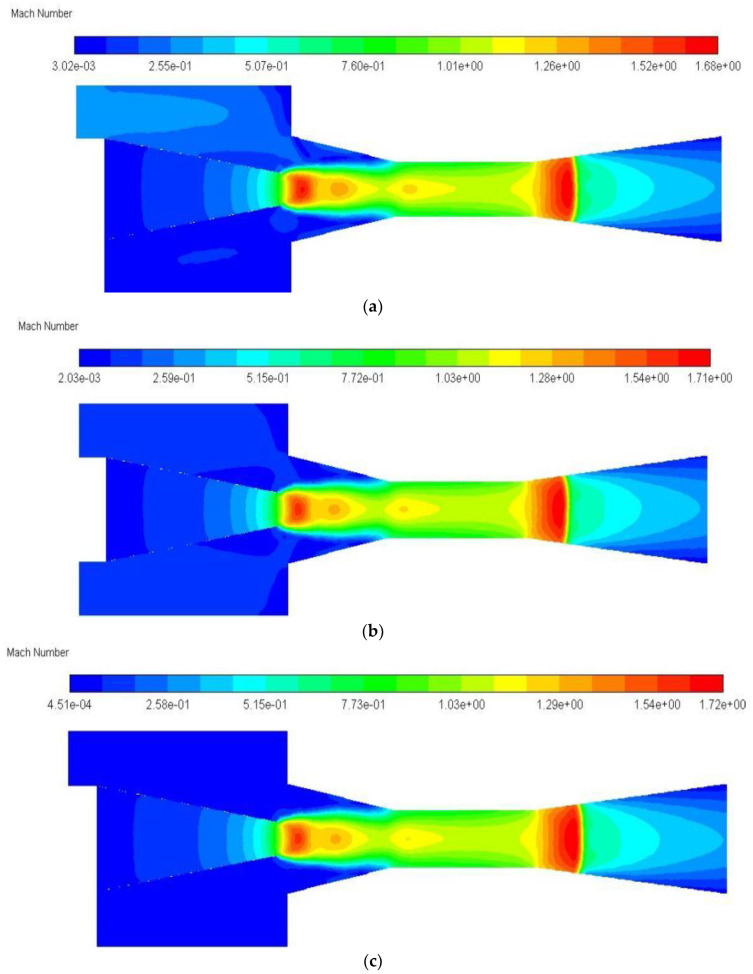
The Mach number distribution in the central symmetry plane (*y* = 0) when secondary flow inlet has: (**a**) one hole; (**b**) two holes; (**c**) five holes; (**d**) six holes; (**e**) seven holes. (Comment: This is the CFD default results for notations).

**Figure 12 entropy-24-01241-f012:**
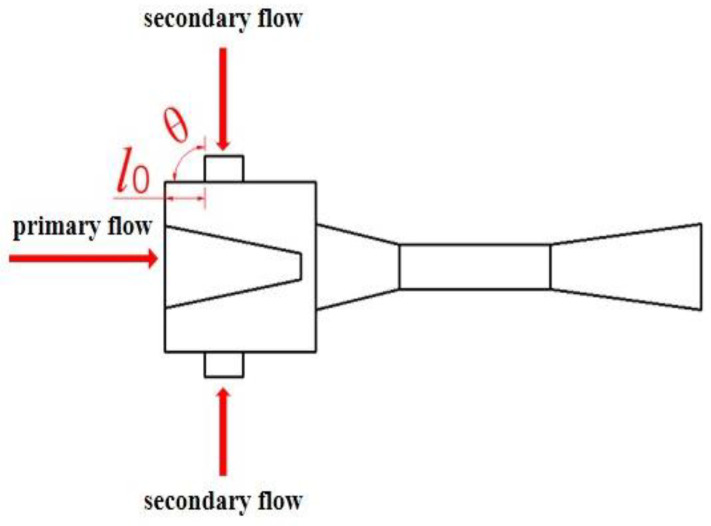
Schematic of the ejector on a symmetric plane (*y* = 0) when the secondary flow inlet is in vertical direction of the primary flow inlet.

**Figure 13 entropy-24-01241-f013:**
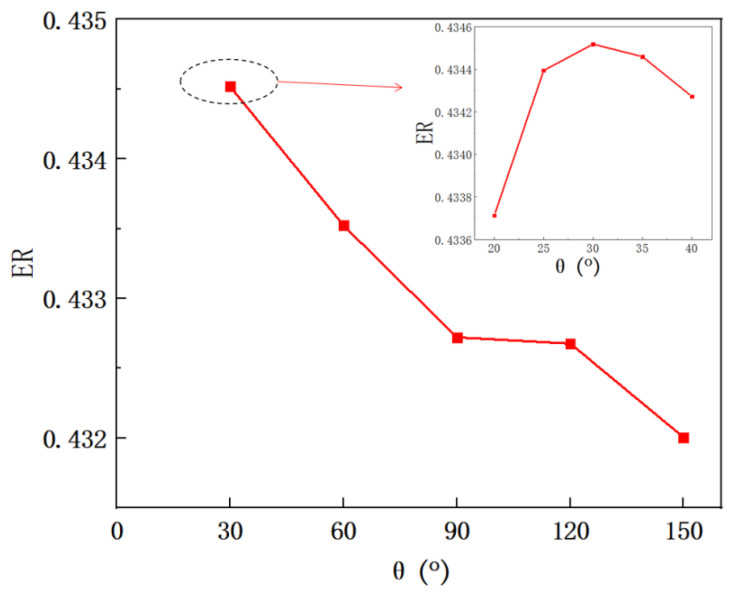
The variation of entrainment ratio with secondary flow inlet angle (the secondary flow inlet is perpendicular to the primary flow inlet).

**Figure 14 entropy-24-01241-f014:**
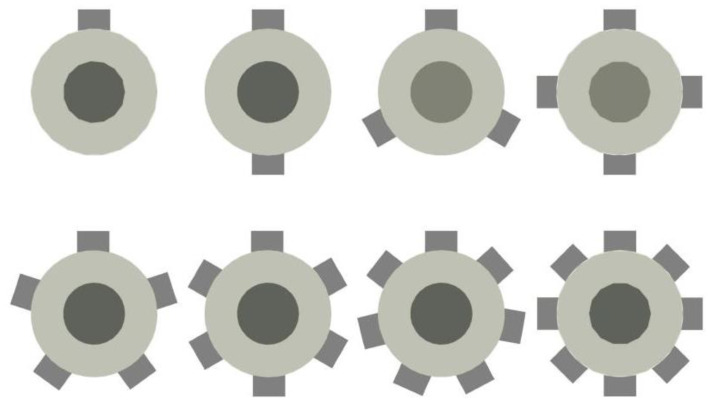
The number and layout of secondary flow inlet holes (the secondary flow inlet is perpendicular to the primary flow inlet).

**Figure 15 entropy-24-01241-f015:**
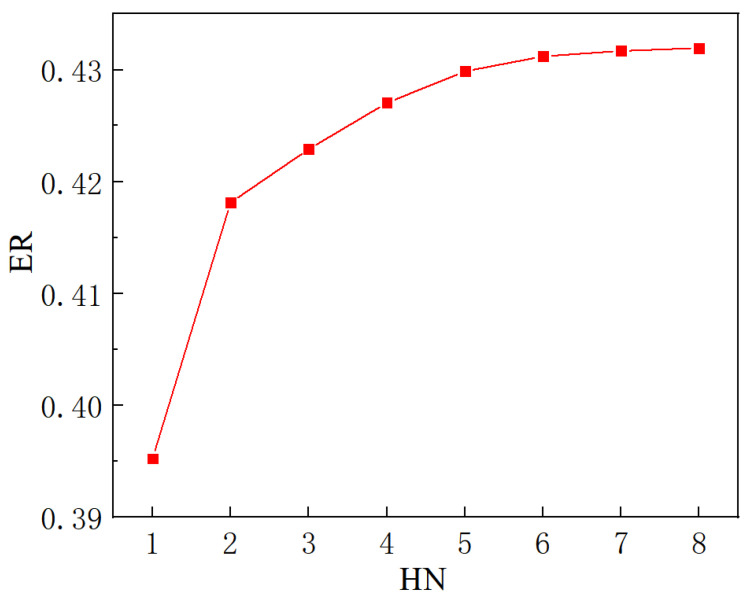
The variation of entrainment ratio with the number of inlet holes of secondary flow (the secondary flow inlet is perpendicular to the primary flow inlet).

**Figure 16 entropy-24-01241-f016:**
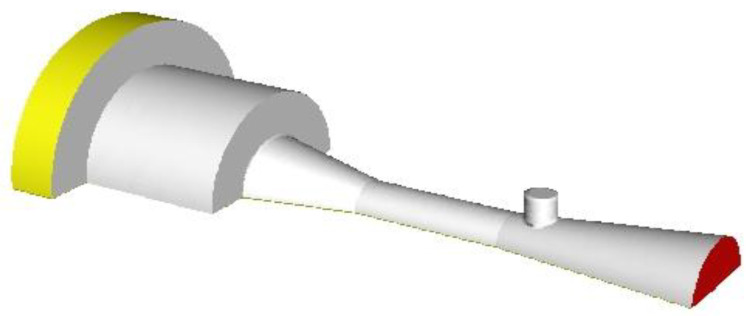
Schematic of an auxiliary circular entrainment hole in the diffuser.

**Figure 17 entropy-24-01241-f017:**
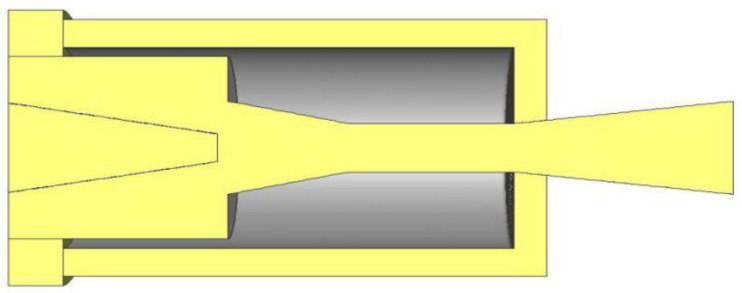
Schematic of the initial piping between auxiliary entrainment inlet and secondary flow inlet.

**Figure 18 entropy-24-01241-f018:**
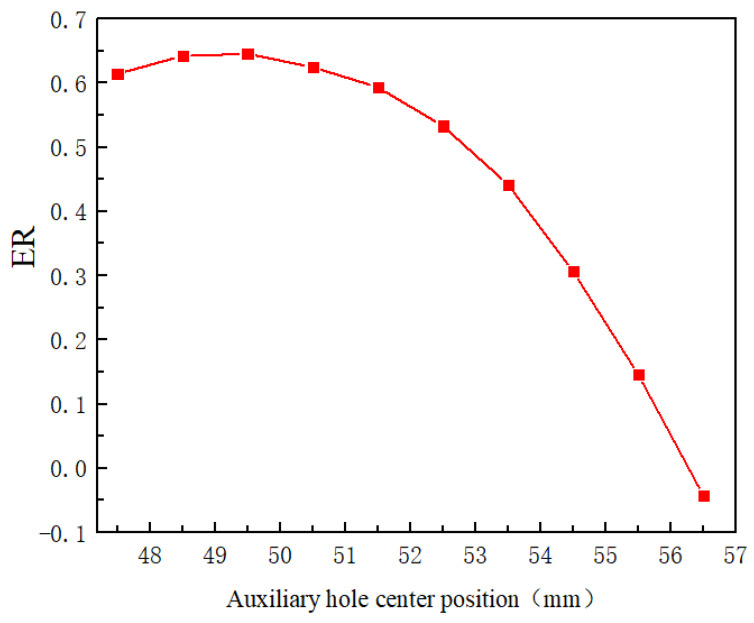
The variation of entrainment ratio with relative position of auxiliary entrainment hole.

**Figure 19 entropy-24-01241-f019:**
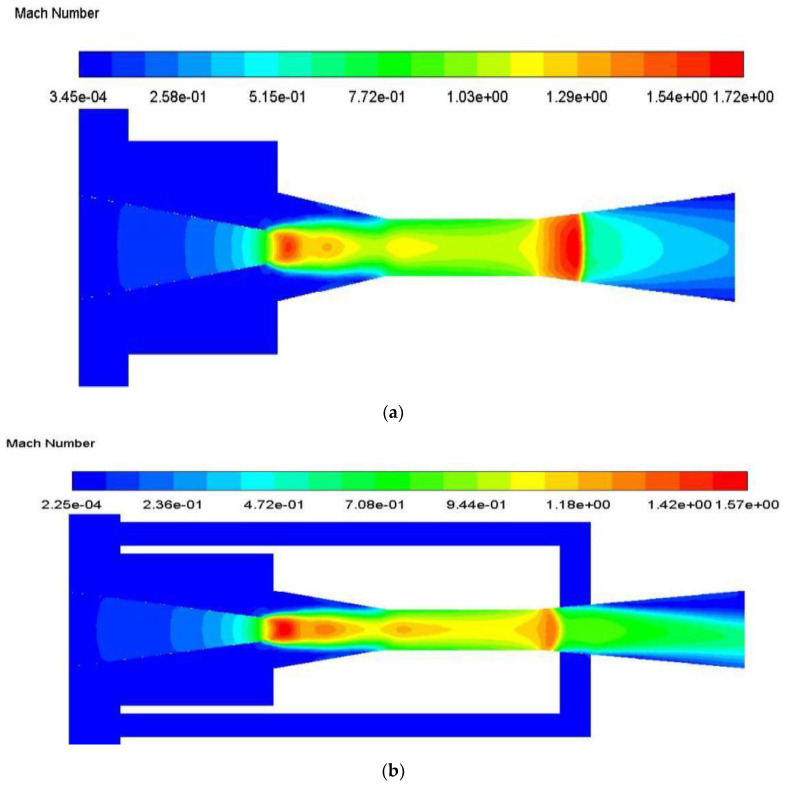
The Mach number distribution of central symmetry plane (*y* = 0) with or without auxiliary entrainment in the diffuser part: (**a**) without auxiliary entrainment; (**b**) with auxiliary entrainment. (Comment: This is the CFD default results for notations).

**Figure 20 entropy-24-01241-f020:**
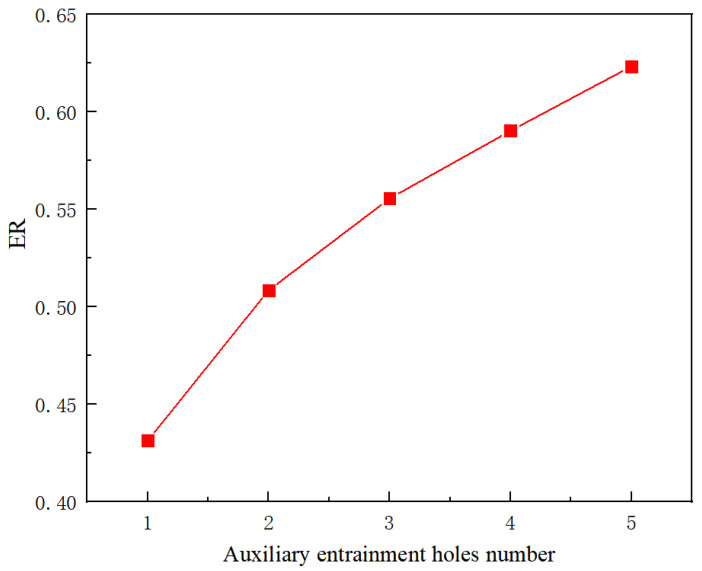
The variation of the entrainment ratio with the number of auxiliary entrainment inlet rectangles under constant axial width.

**Figure 21 entropy-24-01241-f021:**
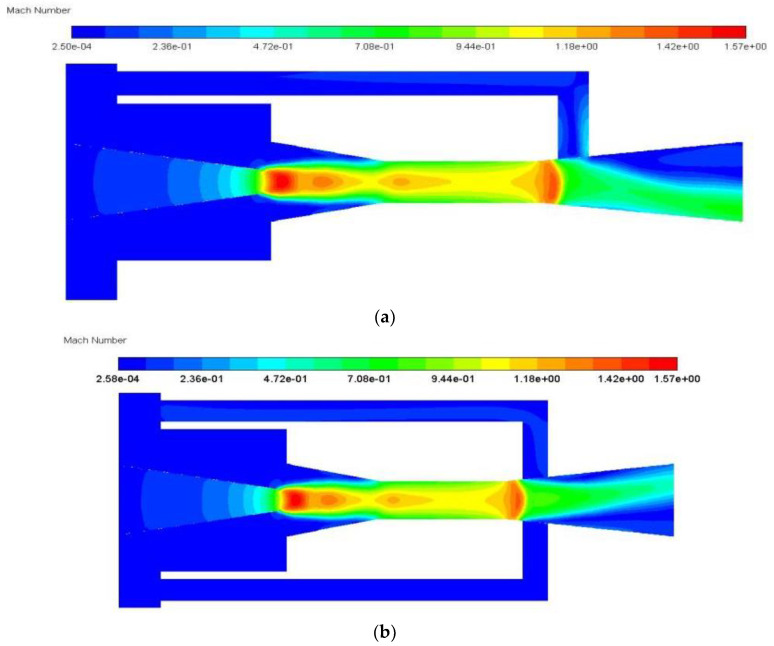
The Mach number distribution of central symmetry plane (*y* = 0) with one rectangle or two symmetric rectangles of auxiliary entrainment in the diffuser part: (**a**) one rectangle; (**b**) two rectangles. (Comment: This is the CFD default results for notations).

**Figure 22 entropy-24-01241-f022:**
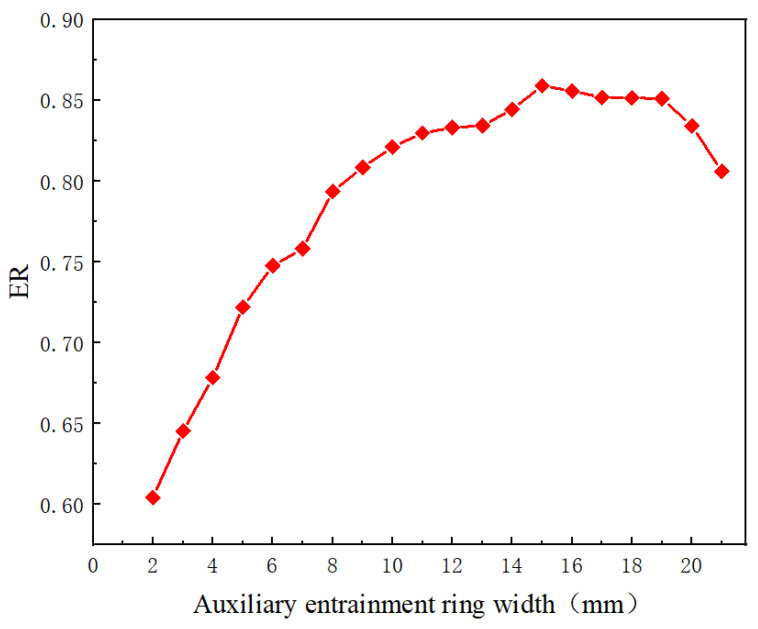
The variation of entrainment ratio with the axial width of auxiliary entrainment inlet ring.

**Figure 23 entropy-24-01241-f023:**
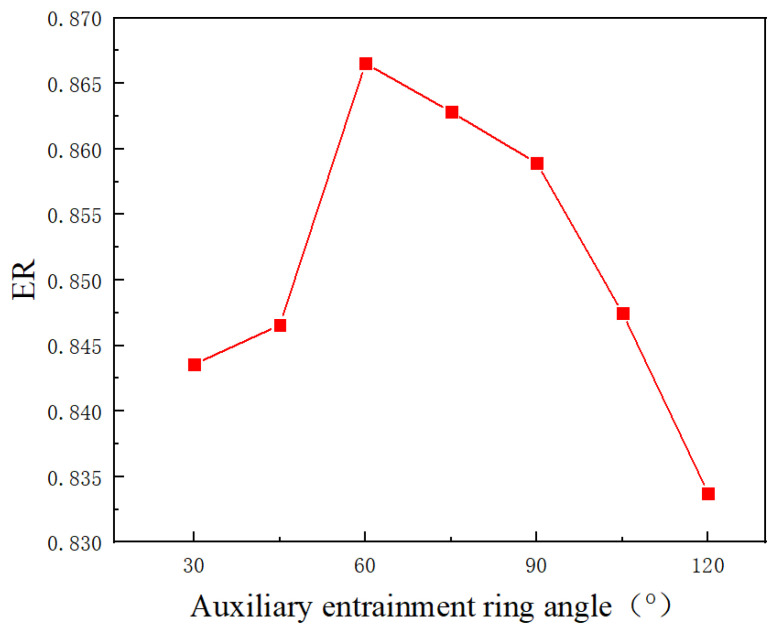
The variation of entrainment ratio with the auxiliary entrainment inlet ring angle.

**Table 1 entropy-24-01241-t001:** Evaporating temperature and designed cooling load of three evaporators.

Evaporators	Evaporating Temperature (°C)	Required Cooling Loads (W)
Evaporator 1 (freezer)	−30	830
Evaporator 2 (refrigerator)	−5	500
Evaporator 3 (air conditioner)	7	1500

**Table 2 entropy-24-01241-t002:** Details of measurement sensors.

Sensors	Type	Unit	Accuracy
Temperature sensors	PT1000	°C	±0.3
Pressure transmitters	Resistance strain	Bar	±0.5%
Volume flow meters	Elliptical gear	L/h	±1.6%

**Table 3 entropy-24-01241-t003:** The geometrical parameters of the initial ejector.

Parameter	Symbol	Value (mm)
Primary nozzle inlet diameter	*D_p_*	9.6
Primary nozzle outlet diameter	*D_n_*	3.0
Secondary inlet diameter	*D_se_*	5.0
Suction chamber diameter	*D_S_*	19.6
Constant-area mixing chamber diameter	*D_am_*	5.2
Diffuser outlet diameter	*D_d_*	10.0
Length of primary nozzle	*L_p_*	18
Length of suction chamber	*L_S_*	20
Length of constant-pressure mixing chamber	*L_pm_*	11
Length of constant-area mixing chamber	*L_am_*	15
Length of diffuser	*L_d_*	20

**Table 4 entropy-24-01241-t004:** Boundary conditions for the ejector.

Parameters	Boundary Condition	Pressure Values (kPa)	Temperature Values (K)
Primary flow	Pressure inlet	243.3	278
Secondary flow	Pressure inlet	84.4	253
Outlet flow	Pressure outlet	92.88	255

**Table 5 entropy-24-01241-t005:** Mesh sensitivity analysis (Point A).

Grid Number	Pressure (kPa)	Error (%)	Velocity (m·s^−1^)	Error (%)
294,275	59.669	/	182.681	/
544,473	61.064	2.34	181.507	−0.64
886,798	61.968	1.48	182.330	0.45

**Table 6 entropy-24-01241-t006:** Twelve sets of specific operating conditions used for experimental validation of the CFD model.

Groups	Primary Flow		Secondary Flow		Back Pressure
	Pressure (kPa)	Temperature (°C)	Pressure (kPa)	Temperature (°C)	Pressure (kPa)
Group 1	262.3, 252.7	−3, −4	84.4	−30	92.8
	234.3, 225.5	−6, −7			
Group 2	243.3	−5	92.7, 88.5,	−28, −29	92.8
			80.4, 76.7,	−31, −32	
Group 3	243.3	−5	84.4	−30	89.46, 94.53
					99.59, 104.66

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
