# Peer review of "Optimization on Secondary Flow and Auxiliary Entrainment Inlets of an Ejector by Using Three-Dimensional Numerical Study"

_entropy, 2022, doi:10.3390/e24091241_

Round 1
Reviewer 1 Report
The paper concerns the three-dimensional numerical simulations and optimization of the ejector.
Comments
page 6, lines 108-111. Please specify what kind of working medium was used on the experimental bench.
page 7, lines 122-123. Please specify the type of sensors used in the measurements performed on the test bench. What is the class of these sensors? Please explain why elliptical gear flowmeters were used.
page 7, Table 2. Please clarify whether the accuracy of the measurement referred to the measuring range of the instrument used or only to the measured value.
page 7, lines 127-140. This section is very weak - please expand it. The determination of measurement uncertainty is very important for experimental work, the results of which are additionally used in numerical modelling. Every measurement of a physical quantity is made with finite accuracy, which means that the result of that measurement is made with measurement uncertainty. This fact is related not only to the imperfection of human actions, but also to the imperfect workmanship of measuring instruments, the random state of matter at the time of measurement, the influence of the measurement process on the measured quantity, and the approximate nature of models of reality described in the form of laws of physics. On what principles the calculation and estimation of measurement uncertainty, as well as the evaluation of the results - please provide the relevant standard.
page 9, lines 157-159. Governing equations must be placed in Section 3.2.
page 9, lines 164-167. Description of the simulation settings must be provided.
REFERENCES The literature should include at least 2 articles on similar topics that were published in the MDPI Entrtopy journal.
Decision - Major Revision
Reviewer 2 Report
In the article, the nomenclature formatting should be corrected, the spaces between paragraphs should be removed, and chapter 2 should not begin with a drawing, it would be good to mention the sources of figures 1, 2 and 3, and table 3 should be entirely on one page. The modelling results are described in detail, but the process itself, modelling assumptions, etc. are abbreviated as in the literature review. This part should be developed.
Round 2
Reviewer 1 Report
The authors have taken into account all the reviewer's comments. The article in the revised version can be published.